# The Low Lying Double-Exciton State of Conjugated Diradicals: Assessment of TDUDFT and Spin-Flip TDDFT Predictions

**Sofia Canola [1] , Yasi Dai [1] and Fabrizia Negri [1,2,*]**

[1]    Department of Chemistry "Giacomo Ciamician", University of Bologna, 40126 Bologna, Italy;
        sofiacanola2@gmail.com (S.C.); daiyasi@hotmail.com (Y.D.)
[2]    INSTM—University of Bologna Research Unit (UdR Bologna), 40126 Bologna, Italy
[*]    Correspondence: fabrizia.negri@unibo.it

**Abstract:** Conjugated singlet ground state diradicals have received remarkable attention owing to their potential applications in optoelectronic devices. A distinctive character of these systems is the location of the double-exciton state, a low lying excited state dominated by the doubly excited HOMO,HOMO→LUMO,LUMO configuration, (where HOMO = highest occupied molecular orbital, LUMO = lowest unoccupied molecular orbital) which may influence optical and other photophysical properties. In this contribution we investigate this specific excited state, for a series of recently synthesized conjugated diradicals, employing time dependent density functional theory (TDDFT) based on the unrestricted parallel spin reference configuration in the spin-flip formulation (SF-TDDFT) and standard TD calculations based on the unrestricted antiparallel spin reference configuration (TDUDFT). The quality of computed results is assessed considering diradical and multiradical descriptors, and the excited state wavefunction composition.

**Keywords:** conjugated diradicals; DFT; broken symmetry; double-exciton state; TDDFT; spin-flip TDDFT; diradical character; $N_{FOD}$ descriptor

## 1. Introduction

Conjugated diradical systems have attracted considerable interest in recent years owing to their potential electronic applications [1] in organic field effect transistors (OFETs) [2–4], organic photodetectors (OPDs) [5], and near-infrared (NIR) dyes [6,7], among others. Significant efforts have been devoted to stabilize the active open-shell molecules, and a large number of stable diradicals with an open-shell singlet ground state have been synthetized with different conjugated cores and varying diradical character [8–12]. There has been a tremendous effort also on the rationalization of the properties from a theoretical point of view, including their linear and non-linear optical properties and their application in singlet fission processes [13], etc.

Although the quantum-chemical description of the singlet ground state of conjugated diradicals calls for multireference methods to include static correlation effects, a very common approach is the use of DFT in its unrestricted formulation (UDFT) which, for the significant diradical character, leads to broken symmetry (BS) molecular orbitals.

A distinctive signature of singlet ground state conjugated diradical systems is the presence of a low lying double-exciton state [14], which in analogy to polyenes—that also display diradical character, especially the longer members [15]—becomes the lowest excited state for large diradical character, as shown in our previous work [16–18]. The double-exciton state is therefore an electronically excited state whose wavefunction is dominated by a large contribution of the HOMO,HOMO→LUMO,LUMO

doubly excited configuration (hereafter indicated as H,H→L,L). The energy location of this excited state can strongly influence the photophysical properties such as linear and non-linear optical properties or fluorescence lifetime of the diradical system, owing to its generally dipole forbidden character [16].

Inclusion of static and dynamic correlation is mandatory for the description of excited states, and CASSCF calculations followed by CASPT2 or NEVPT2 calculations have been employed successfully [8,16,17]. Time dependent (TD) DFT calculations suffer from the lack of inclusion of multiple excitations, however spin-flip (SF) TDDFT includes the H,H→L,L configuration and other multiple excitations, owing to the reference triplet (or parallel-spin) configuration [19,20]. In addition to SF-TDDFT, we have recently shown [18] that also TDUDFT calculations, namely TDDFT calculations based on an unrestricted BS antiparallel-spin reference configuration, can capture double-exciton states dominated by the H,H→L,L configuration. Based on the above considerations, in a previous work [18] we have investigated a sample of eight recently synthesized conjugated diradicals displaying experimental evidence of the double-exciton state and have computed its excitation energy with SF-TDDFT and TDUDFT. SF-TDB3LYP calculations were found to perform better for small to medium diradical character while TDUB3LYP was shown to produce satisfactory results only for large diradical character.

Here, we extend the study and consider the four recently synthesized conjugated diradicals displaying a singlet ground state shown in the top part of Figure 1: Nonazethrene (NZ) [9], superoctazethrene (SOZ) [10], diindenophenanthrene derivative (DIPh) [12], and peri-tetracene (PT) [11]. This set is added to the previous set of molecules for the investigation of the double-exciton state. We use the TDUDFT and SF-TDDFT approaches with the already tested B3LYP functional but for the latter method we also adopt the BHHLYP functional for the entire set of twelve molecules shown in Figure 1. The computed results are compared with experimental data and their quality is assessed by analyzing diradical and multiradical character descriptors along with the wavefunction composition generated by the two approaches.

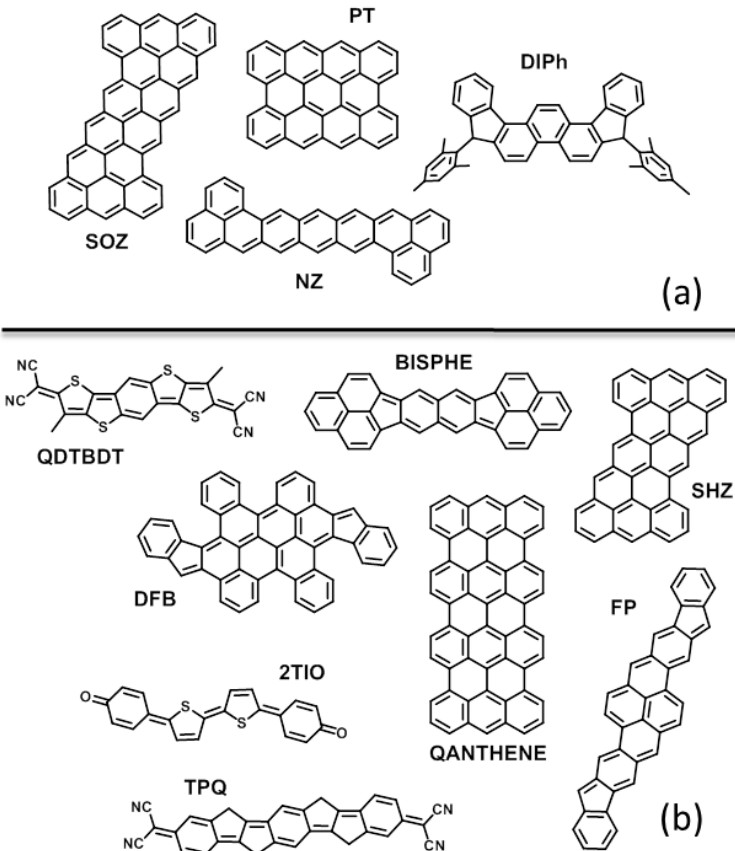

**Figure 1.** Structural formula of the conjugated diradicals (**a**) Nonazethrene (NZ), Superoctazethrene (SOZ), Diindenophenanthrene derivative (DIPh), and Peri-tetracene (PT) investigated in this work for

the first time and (**b**) Thiophene-Based Heterophenoquinone (2TIO), Quinoidal dithieno-benzo-dithiophene (QDTBDT), Bis-phenalenyl (BISPHE), Diindeno-fused bischrysene (DFB), Bis-fluoreno-pyrene (FP), Tri-*p*-quinodimethane (TPQ), Superheptazethrene (SHZ), and Quarteranthene (QANTHENE), considered also in a previous study [18], all displaying an open-shell singlet ground state.

## 2. Computational Details

The equilibrium structures of the molecules shown in Figure 1 were determined with DFT calculations employing the B3LYP functional and the 6-31G* basis set. The geometry optimization was first carried out with the restricted approach to determine a closed-shell (CS) equilibrium structure. For all the systems investigated a more stable open-shell BS solution was found at the CS geometry and therefore the equilibrium structure corresponding to the BS solution was readily determined. The overall CS-BS stability ($\Delta$E(CS-BS)) was determined as the energy difference between the energy of the CS structure computed with restricted DFT and the energy of the BS structure computed with UDFT.

Two descriptors of the diradical/multiradical character were employed. The first is the $y_0$ parameter which, in the spin-unrestricted single-determinant formalism, can be determined in the spin-projection scheme as [13,21]:

$$y_0^{PUnrestricted} = 1 - \frac{2T_0}{1 + T_0^2} \tag{1}$$

with $T_0$ calculated as:

$$T_0 = \frac{n_{HONO} - n_{LUNO}}{2} \tag{2}$$

and $n$ is the occupation number of the frontier natural orbitals.

The second parameter considered is based on finite-temperature DFT (FT-DFT) and is the $N_{FOD}$ value, which is the integral of the fractional orbital density (FOD) $\rho^{FOD}(r)$, over all space:

$$N_{FOD} = \int \rho^{FOD}(r)dr \tag{3}$$

The $\rho^{FOD}(r)$ is defined as [22,23]:

$$\rho^{FOD}(r) = \sum_i^N (\delta_1 - \delta_2 f_i)|\varphi_i(r)|^2 \tag{4}$$

where $\delta_1$ and $\delta_2$ are two constants set such that only fractionally occupied orbitals are taken into account; $\varphi_i$ are molecular spin orbitals, and $f_i$ are the fractional orbital occupancies ($0 \leq f_i \leq 1$) determined by the Fermi-Dirac distribution. In other words, the so defined FOD yields, for each point in real space, only the contribution of 'hot' or strongly correlated electrons and is therefore an analysis tool of static correlation. The $y_0$ and $N_{FOD}$ parameters were computed at the BS geometry for the entire set of diradicals shown in Figure 1. The $N_{FOD}$ parameter was computed with the ORCA 4.0.1.2 package [24] with the default setting (TPSS/def2-TZVP level with $T_{el}$ = 5000 K).

The simplest quantum-chemical model to describe a diradical includes two electrons in two orbitals (2e-2o) [14]. The double-exciton state emerges as one of the two singlet excited states from a full configuration interaction (CI) within the 2e-2o model. A reliable prediction of its excitation energy is however challenging because of correlation effects and generally MCSCF + CASPT2 or similarly correlated methods are required. However, in a recent work we have shown that, for systems with well localized BS frontier MOs, TDUDFT calculations can be used to predict the excitation energy of the double-exciton state, besides that of the single-exciton state, since both excited states are described in terms of singly excited configurations [18]. Double excitations can be recovered from TDDFT calculations also with the SF scheme [19,20]. Spin-flipping excitations enable SF-TDDFT to

treat ground- and excited-state electron correlation on the same footing, while also incorporating some doubly excited determinants that are important for biradicals [19]. Accordingly, these two approaches were employed to investigate the excitation energy of the low lying double-exciton state of the molecules shown in Figure 1.

Geometry optimization and TDUB3LYP calculations were carried out with the Gaussian16 suite of programs [25] while SF-TDDFT calculations were carried out in the collinear approximation as implemented in the GAMESS package [26].

## 3. Results

### 3.1. Stability of the BS Structures and Descriptors of Di/Multiradical Character

All the systems in Figure 1 display a more stable BS structure (see Table 1 and the supplementary file accompanying this work (SFAW)) with PT, TPQ, SHZ, SOZ, and QANTHENE showing the larger energy difference between CS and BS structures. The computed stabilities of the open-shell structures correlate closely with the values of the diradical character $y_0$, as shown in Figure 2. The molecules displaying the larger $y_0$ are also those displaying the larger BS-CS stabilization energy and, to a good approximation, a linear correlation is found between the diradical character and computed BS stabilization for the entire set of twelve diradicals (see Figure 2). The increasing stabilization of the BS structure is accompanied by an increase in localization of the BS frontier orbitals as documented by the graphical representations in Figure 3 for PT and SOZ and in SFAW for the other two systems investigated, and by the computed overlap between BS frontier orbitals pairs $H_\alpha, H_\beta$ and $L_\alpha, L_\beta$ (see SFAW), whose value decreases with increasing localization.

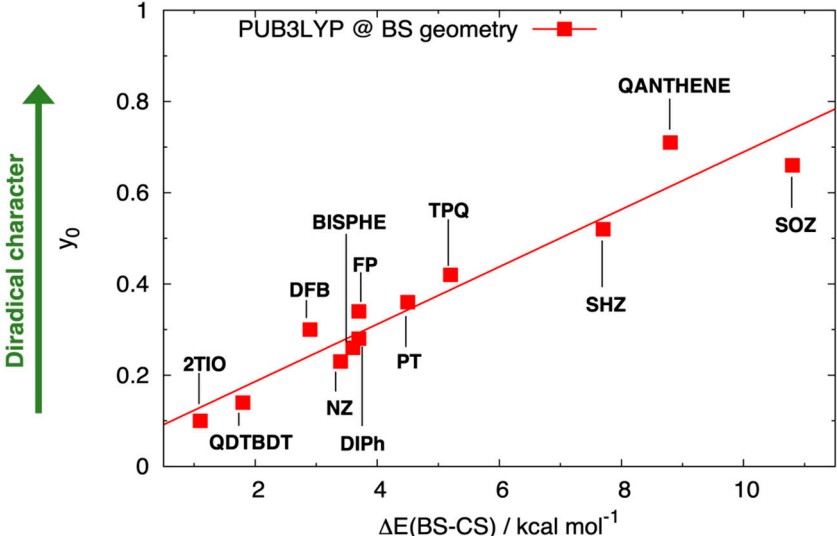

**Figure 2.** Correlation between the $y_0$ value computed at projected UB3LYP level PUB3LYP (red squares) and the computed stabilization of the BS structure with respect to the closed-shell (CS) structure (Energy(CS)-Energy(BS)), both optimized at B3LYP/6-31G* level. Some data are taken from [18]. The coefficient of determination R-squared($R^2$) value is 0.93.

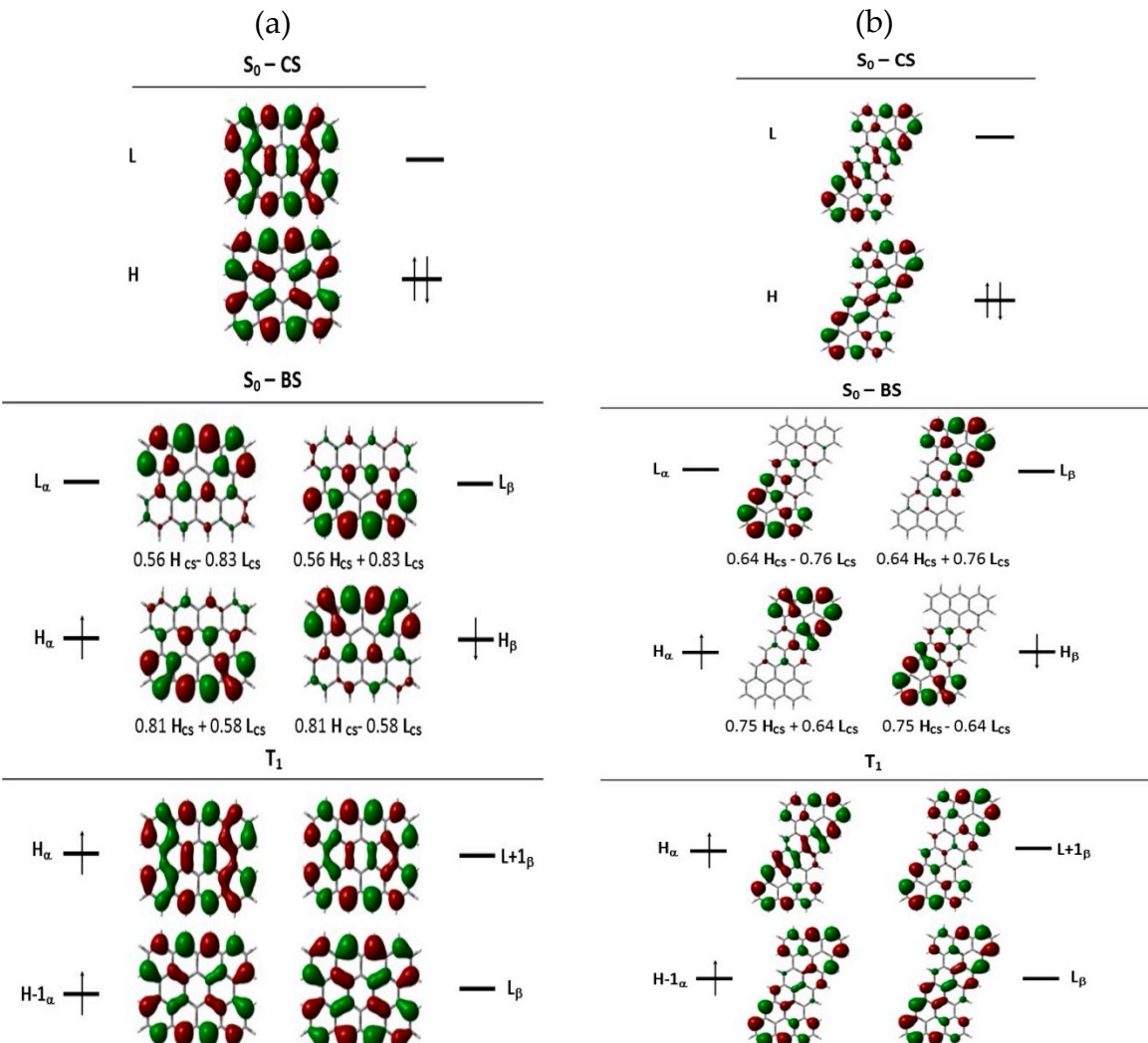

**Figure 3.** Frontier molecular orbitals of (**a**) PT and (**b**) SOZ computed with (top) a CS singlet reference configuration, (middle) a BS singlet open-shell configuration, and (bottom) a triplet configuration, at the optimized BS UB3LYP geometry of the singlet ground state ($y_0$(PUB3LYP) = 0.36 for PT and 0.66 for SOZ). Each BS orbital is also expressed as a linear combination of the delocalized (CS) orbitals.

**Table 1.** Computed values of the descriptors $N_{FOD}$ and $y_0(PUB3LYP)$ and stability of the BS structure with respect to the CS structure, computed at B3LYP/6-31G* level.

| Molecule | $N_{FOD}$ | $y_0(PUB3LYP)$ | $\Delta E$(BS-CS)/kcal mol$^{-1}$ |
|:---:|:---:|:---:|:---:|
| 2TIO | 1.60 | 0.10 [1] | 1.1 [1] |
| QDTBDT | 1.55 | 0.14 [1] | 1.8 [1] |
| NZ | 1.80 | 0.23 | 3.4 |
| BISPHE | 1.70 | 0.26 [1] | 3.6 [1] |
| DIPh | 1.75 | 0.28 | 3.7 |
| DFB | 2.26 | 0.30 [1] | 2.9 [1] |
| FP | 1.90 | 0.34 [1] | 3.7 [1] |
| PT | 1.87 | 0.36 | 4.5 |
| TPQ | 1.75 | 0.42 [1] | 5.2 [1] |
| SHZ | 2.14 | 0.52 [1] | 7.7 [1] |
| SOZ | 2.58 | 0.66 | 10.8 |
| QANTHENE | 2.34 | 0.71 [1] | 8.8 [1] |

[1] From [18].

In previous works, the $N_{FOD}$ values have been compared with the $y_0$ values for linear acenes [27] and have been used to demonstrate the poly-radical character of cyclacenes [27] and single-wall carbon nanotubes [28].

The combination of the two descriptors gives therefore complementing information on the reliability of the simple 2e-2o approach to describe a diradical system. For the sample of diradicals investigated here and in previous work [18] it can be seen that the $N_{FOD}$ value (Table 1) remains below 2 in most cases, indicating that the 2e-2o model should be suitable, but in some cases it exceeds 2, such as for DFB, SHZ, SOZ, and QANTHENE. In these cases it is expected that more than two electrons should be correlated for a proper description of ground and excited states.

### 3.2. Excitation Energies of the Double-Exciton State

For the systems in Figure 1, we have carried out two sets of TDDFT calculations to determine the location of the double-exciton state: (1) TDUDFT calculations using a BS reference configuration at the BS optimized geometry, (2) SF-TDDFT calculations carried out with a triplet reference configuration at the same BS optimized geometry as above. The B3LYP functional was selected for both these calculations owing to its generally good performance for large conjugated systems [29–31] and its generally smaller spin contamination compared to hybrid functionals with larger contributions of HF exchange [32]. However, because the SF calculations are run in the collinear approximation, a functional including a larger amount of exchange correlation is recommended [19,20] and therefore we run the SF-TDDFT calculations also with the BHHLYP functional.

The set of new computed excitation energies in Table 2 together with those obtained in previous work are used in Figure 4 to graphically represent the accuracy of TDUB3LYP and SF-TDBHHLYP calculations. To easily capture a qualitative trend in the computed results, we also show a linear fitting of the computed data. SF-TDBHHLYP predicts excitation energies in average good agreement with the observed values although the fitting line indicates a general underestimate. Notably, larger $y_0$ values are associated with slightly larger underestimates, possibly due to an imperfect description of static correlation effects for very large diradical character. The accuracy of SF-TDBHHLYP calculations is also more satisfactory than that of SF-TDB3LYP (see SFAW). The present results confirm, as already noted [18] that the accuracy of TDUB3LYP is generally acceptable for large $y_0$ values although an increase of the error in predicted excitation energies is documented also for increasingly larger $y_0$ values (see the "discussion section" for more details). The method is generally not reliable for $y_0$ (PUB3LYP) lower than 0.3.

**Table 2.** Comparison between time dependent unrestricted density functional theory (TDUDFT) and spin-flip (SF-TDDFT) computed excitation energies and experimental values (eV) of the lowest energy double-exciton state for the systems investigated in this work.

| Molecule | TDUB3LYP 6-31G* | SF-TDB3LYP 6-31G* | SF-TDBHHLYP 6-31G* | exp. |
|---|---|---|---|---|
| **2TIO** | 0.98 [1] | 1.54 [1] | 1.35 | 1.68 [2] |
| **QDTBDT** | 1.07 [1] | 1.56 [1] | 1.56 | 1.57 [3] |
| **NZ** | 1.22 | 1.60 | 1.16 | 1.39 [4] |
| **BISPHE** | 1.03 [1] | 1.27 [1] | 1.57 | 1.54 [5] |
| **DIPh** | 1.21 | 1.17 | 1.23 | 1.18 [6] |
| **DFB** | 0.91 [1] | 0.86 [1] | 0.88 | 0.92 [7] |
| **FP** | 1.13 [1] | 1.05 [1] | 0.93 | 1.13 [8] |
| **PT** | 1.13 | 1.09 | 1.56 | 1.23 [9] |
| **TPQ** | 1.16 [1] | 1.04 [1] | 1.01 | 1.13 [10] |
| **SHZ** | 1.32 [1] | 0.98 [1] | 1.12 | 1.19 [11] |
| **SOZ** | 1.41 | 1.00 | 0.79 | 1.07 [12] |
| **QANTHENE** | 1.17 [1] | 0.76 [1] | 0.94 | 1.08 [13] |

[1] From [18], [2] From [16], [3] From [33], [4] From [9], [5] From [34], [6] From [12], [7] From [35], [8] From [36], [9] From [11], [10] From [37], [11] From [38], [12] From [10], [13] From [8].

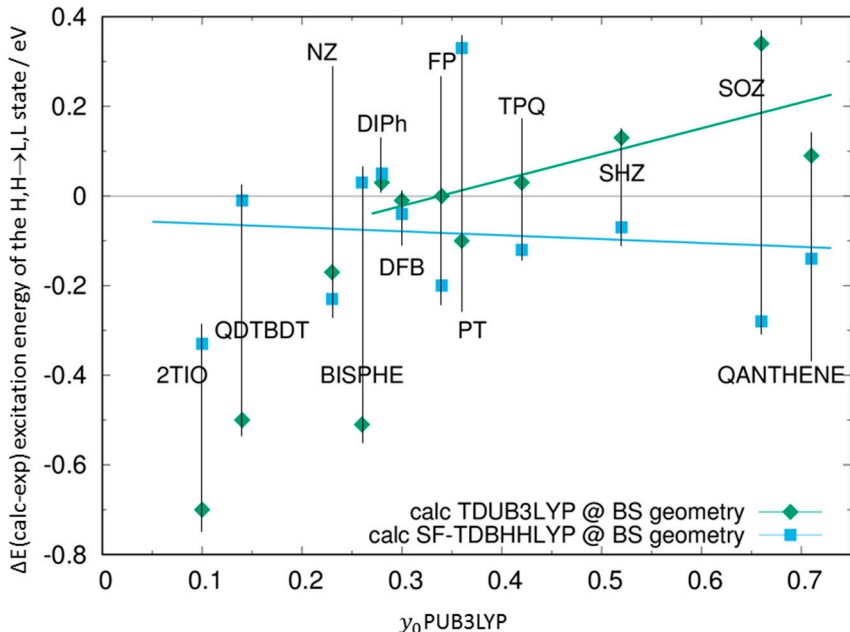

**Figure 4.** Difference between computed and observed excitation energy of the double-exciton (H,H→L,L) state versus the computed $y_0$ at PUB3LYP level: (Light blue squares) SF-TDBHHLYP at BS B3LYP geometry; (green diamonds) TDUB3LYP at the same geometry. The lines in the same colours are linear fittings of the computed data. Vertical bars indicate the compound to which computed data correspond. Some of the TDUB3LYP results are taken from [18].

## 4. Discussion

The results of TDUDFT and SF-TDDFT calculations can be critically analyzed by considering (a) the diradical character and the reliability of the 2e-2o model for the systems investigated; (b) the derivation of the wavefunction describing the doubly excited state at TDUDFT level when the 2e-2o model holds; (c) the indication provided by $N_{FOD}$ parameter, and (d) the role of spin-contamination.

First, we consider how the localization of BS orbitals impacts the diradical character $y_0$. Within the 2e-2o model, the BS HOMO and LUMO orbitals of the unrestricted wavefunction can be described as linear combinations of the delocalized $H_{CS}$ and $L_{CS}$ orbitals obtained from the CS solution. Following previous work [13,21], we can write:

$$H_\alpha = \varphi^\alpha_{HOMO} = \cos\theta H_{CS} + \sin\theta L_{CS} \tag{5}$$

$$H_\beta = \varphi^\beta_{HOMO} = \cos\theta H_{CS} - \sin\theta L_{CS} \tag{6}$$

$$L_\alpha = \varphi^\alpha_{LUMO} = \sin\theta H_{CS} - \cos\theta L_{CS} \tag{7}$$

$$L_\beta = \varphi^\beta_{LUMO} = \sin\theta H_{CS} + \cos\theta L_{CS} \tag{8}$$

where $\theta$ is the angle of rotation with respect to the CS set of orbitals. By substituting the expressions (5–8) in the Slater determinant corresponding to the unrestricted ground state wavefunction:

$$\psi^{Unrestricted} = \left|\varphi^\alpha_{HOMO}\varphi^\beta_{HOMO}\right\rangle = \frac{1}{\sqrt{2}}\left[\varphi^\alpha_{HOMO}(1)\varphi^\beta_{HOMO}(2) - \varphi^\beta_{HOMO}(1)\varphi^\alpha_{HOMO}(2)\right] \tag{9}$$

the wavefunction describing the singlet open-shell ground state can be expressed as:

$$\begin{aligned}\psi^{Unrestricted} &= \left(\cos^2\theta\right)\left|H_{CS}\overline{H}_{CS}\right\rangle - \left(\sin^2\theta\right)\left|L_{CS}\overline{L}_{CS}\right\rangle \\ &\quad - \sin\theta\cos\theta\left(\left|H_{CS}\overline{L}_{CS}\right\rangle - \left|L_{CS}\overline{H}_{CS}\right\rangle\right)\end{aligned} \tag{10}$$

or, in terms of combination coefficients $C_{GR}, C_D, C_T$:

$$\psi^{Unrestricted} = C_{GR}\left|H_{CS}\overline{H}_{CS}\right\rangle + C_D\left|L_{CS}\overline{L}_{CS}\right\rangle + C_T\frac{1}{\sqrt{2}}\left(\left|H_{CS}\overline{L}_{CS}\right\rangle - \left|L_{CS}\overline{H}_{CS}\right\rangle\right) \tag{11}$$

where $\left|H_{CS}\overline{H}_{CS}\right\rangle$ is the CS ground state determinant, $\left|L_{CS}\overline{L}_{CS}\right\rangle$ is the doubly excited determinant and $\frac{1}{\sqrt{2}}(|H_{CS}\overline{L}_{CS}\rangle - |L_{CS}\overline{H}_{CS}\rangle)$ is the combination of singly excited determinants corresponding to a triplet spin multiplicity, which accounts for spin contamination.

The $y_0$ parameter can be expressed in terms of the doubly excited configuration and triplet state contributions (see Equation (11)) as [13]:

$$y_0 = 2(C_D)^2/(1 - (C_T)^2) \tag{12}$$

from which the following dependence on the CS to BS orbital rotation angle $\theta$ can be deduced, recalling Equation (10):

$$y_0 = 2(\sin^4\theta)/(1 - 2(\sin^2\theta)(\cos^2\theta)). \tag{13}$$

The linear combination of the BS frontier orbitals of each molecule in Figure 1 is determined by projecting each BS frontier orbital over the set of CS orbitals. The corresponding combination coefficients, from which the $\theta$ values (see also SFAW) can be determined, are collected in Figure 3 for PT and SOZ, in SFAW for the other molecules investigated here, and in [18] for those previously investigated. For the entire set of molecules investigated, the $\theta$ values determined either from the expression of the BS HOMO in terms of CS orbitals or from the expression of the BS LUMO in terms of the CS orbitals, are depicted as green crosses and blue crosses, respectively, in Figure 5, where Equation (13) is also plotted (red curve).

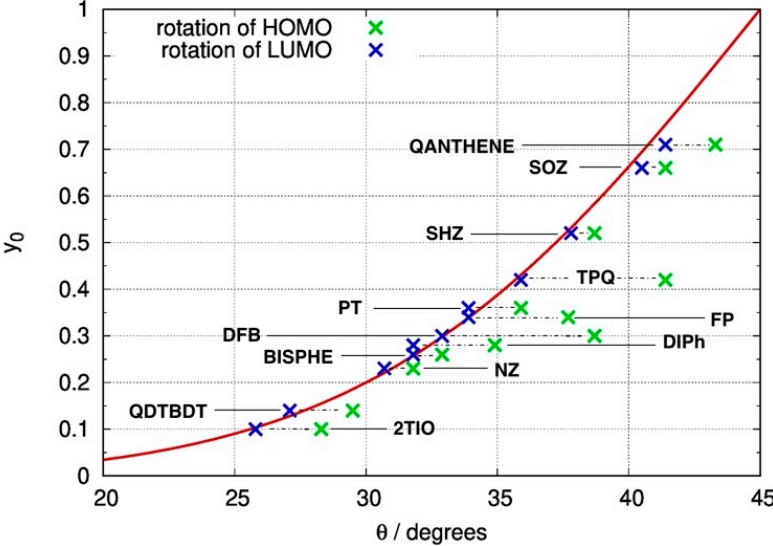

**Figure 5.** Dependence of the diradical character $y_0$(PUB3LYP) as a function of BS orbital rotation angle $\theta$. Analytic expression from Equation (13) (red curve); $\theta$ determined for the HOMO BS orbitals (green crosses); $\theta$ determined for the LUMO BS orbitals (blue crosses).

There is a general good agreement between the orbital rotation angle determined from the BS and CS orbitals of each molecule and the theoretical red curve, which suggests that orbital rotation, especially for the BS LUMO orbital, can be taken as an additional descriptor of the diradical character. Clearly, an orbital rotation of 45° implies the formation of fully localized BS orbitals and corresponds to the maximum diradical character $y_0 = 1$. When the rotation angle is smaller, $y_0$ decreases and becomes zero when CS and BS orbitals coincide. The largest discrepancies observed for the green crosses

(rotation angle deduced from the expression of the BS HOMO orbital) mainly arise from additional contributions to the BS orbital from other CS occupied orbitals (HOMO-1 and lower). Such extra contributions suggest that in these cases the 2e-2o model may not fully apply.

We now consider a derivation of the wavefunction describing the double-exciton state for the TDUDFT approach. The double-exciton state is identified by the positive combination of the two single excitations $[H_\alpha \to L_\alpha] + [H_\beta \to L_\beta]$ as discussed in detail in [18] or, in terms of the corresponding Slater determinants, as:

$$
\begin{aligned}
\psi^{Double\ exciton\ state} &= \frac{1}{\sqrt{2}}\left\{\left|\varphi^\alpha_{LUMO}\varphi^\beta_{HOMO}\right\rangle + \left|\varphi^\alpha_{HOMO}\varphi^\beta_{LUMO}\right\rangle\right\} \\
&= \frac{1}{2}\left[\varphi^\alpha_{LUMO}(1)\varphi^\beta_{HOMO}(2) - \varphi^\beta_{HOMO}(1)\varphi^\alpha_{LUMO}(2)\right] \\
&= \frac{1}{2}\left[\varphi^\alpha_{HOMO}(1)\varphi^\beta_{LUMO}(2) - \varphi^\beta_{LUMO}(1)\varphi^\alpha_{HOMO}(2)\right]
\end{aligned}
\tag{14}
$$

substituting the expressions (5–8) into (14) one gets:

$$
\begin{aligned}
\psi^{Double\ exciton\ state} &= \frac{1}{\sqrt{2}}\left\{2\sin\theta\cos\theta\left|H_{CS}\overline{H}_{CS}\right\rangle + 2\sin\theta\cos\theta\left|L_{CS}\overline{L}_{CS}\right\rangle\right. \\
&\left. + \left(\cos^2\theta - \sin^2\theta\right)\left(\left|H_{CS}\overline{L}_{CS}\right\rangle - \left|L_{CS}\overline{H}_{CS}\right\rangle\right)\right\}
\end{aligned}
\tag{15}
$$

or

$$
\psi^{Double\ exciton\ state} = D_{GR}\left|H_{CS}\overline{H}_{CS}\right\rangle + D_D\left|L_{CS}\overline{L}_{CS}\right\rangle + D_T\frac{1}{\sqrt{2}}\left(\left|H_{CS}\overline{L}_{CS}\right\rangle - \left|L_{CS}\overline{H}_{CS}\right\rangle\right)
\tag{16}
$$

where the combination coefficients can also be recast as $D_{GR} = D_D = \sin(2\theta)/\sqrt{2}$ and $D_T = \cos(2\theta)$. Equation (16) shows that the wavefunction of the double-exciton state, as in the case of the ground state, is composed by the combination of singlet and triplet spin contributions which implies that spin contamination can be a relevant issue. In Figure 6a, we see that for angles increasing from 0 to 45° the square of the coefficient of the triplet spin multiplicity component $(D_T)^2$ decreases from 1 to 0, while the sum of the two singlet spin multiplicity components, $(D_{GR})^2 + (D_D)^2$, increases from 0 to 1. This implies that the double-exciton state wavefunction is a pure singlet state only for fully localized BS orbitals, while decreasing the $\theta$ rotation angle, the triplet component becomes more and more important and for $\theta = 0$ the state becomes a pure triplet state. Since we have just shown how $\theta$ is related to the diradical character, the above results demonstrate that for very small rotation angles (and diradical character) the triplet contribution dominates and the predicted TDUDFT result is unreliable because of the mixed spin nature of the state which implies a large spin contamination. As shown in Figure 6b, all the molecules investigated display a $\theta$ larger than 20° but in some cases the contribution from the triplet component (the blue crosses in Figure 6b) is larger than 20%, which results in a non-negligible spin-contamination of the predicted excited state.

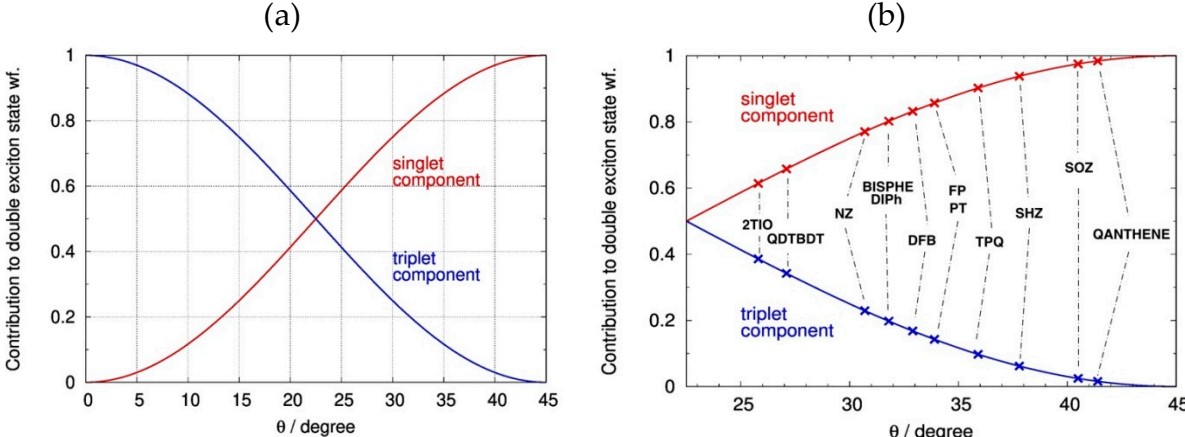

**Figure 6.** Contributions of the singlet and triplet components to the double-exciton state described at TDUDFT level according to Equation (15) and (16) as a function of BS orbital rotation angle $\theta$. (**a**) Contributions over the entire range of $\theta$ values; (**b**) enlargement over the range of $\theta$ values computed for the twelve diradicals investigated.

Based on the above analysis it is clear that the TDUDFT approach can be used to describe the double-exciton state when (a) the 2e-2o model is valid to a good extent and (b) when the diradical character is large, otherwise mixing with the triplet component and spin contamination that invalidate the results. This explains the general inaccuracy of the method for molecules displaying $y_0$(PUB3LYP) smaller than ca. 0.3 in Figure 4. There is, however, an additional limit emerging from Equation (15), namely the identical contribution of ground $\left|H_{CS}\overline{H}_{CS}\right\rangle$ and doubly excited $\left|L_{CS}\overline{L}_{CS}\right\rangle$ configurations to the singlet state component provided by the TDUDFT description. This restriction may not be optimal and may lead to inconsistencies also for systems displaying marked diradical character. Finally, additional inaccuracies may be originated by the presence of a multiradical character which can be monitored by considering the $N_{FOD}$ descriptor. As noted above, the $N_{FOD}$ indicates that more than two electrons need to be correlated in some cases (see Table 1). The last two observations explain some larger deviations of TDUB3LYP results from the experiment, documented in Figure 4 also for $y_0$ above 0.6, for SHZ, SOZ, and QANTHENE. In summary, the TDUB3LYP may provide a good estimate of the energy location of the double-exciton state only when several conditions hold at the same time as discussed above.

Concerning the SF method, the results shown in Figure 4 are generally of similar quality for all $y_0$ values because the SF wavefunction does not suffer from the contamination with the awkward triplet component of the TDUDFT approach shown in Equation (15). Despite the favorable features of SF-TDDFT, only those excitations within the open-shell space are able to generate spin-pure solutions, whereas all other configurations are missing their "spin complements", leading to spin-contaminated solutions [19]. Therefore, spin contamination may affect the quality of the results owing to the spin incomplete expansion of the wavefunction beyond the 2e-2o orbital space. In this regard the SF wavefunction is suitable to describe systems displaying also some degree of multiradical character because double excitations are not limited to the 2e-2o space, although some double excitations outside this space are missing.

In light of the above discussion, the origin for the scattered results predicted for some of the systems investigated, can be attributed, for TDUDFT calculations, to (a) small diradical character, (b) deviation from the 2e-2o model as documented by the large $N_{FOD}$ value, (c) and excited state requiring a wavefunction in which the contribution of ground and doubly excited configurations should not be forced to be identical. For SF-TDDFT calculations, the few scattered results are mainly attributed to a) spin contamination and b) lack of some relevant double excitations in the SF wavefunction. We can conclude that SF-TDDFT is suitable to describe the double-exciton state for a more extended range of diradical systems compared to TDUDFT.

## 5. Conclusions

A distinctive signature of singlet ground state conjugated diradical systems is the presence of a low lying double-exciton state. We have investigated this excited state employing TDDFT based on the unrestricted parallel spin reference configuration in the spin-flip formulation (SF-TDDFT) and standard TD calculations based on the unrestricted antiparallel spin reference configuration (TDUDFT).

For all the systems investigated we have determined the rotation angle $\theta$ between the BS and CS frontier orbitals and shown that the computed diradical character $y_0$ follows the expected dependence on $\theta$.

Based on the analysis of the wavefunction describing the double-exciton state at the TDUDFT level, we have shown that only for large diradical character the triplet state contribution is minor. However, the contribution of the ground state and doubly excited configurations remains identical for every value of $y_0$. When these restrictions hold, and the 2e-2o model represents the system well, a good estimate of the energy location of the double-exciton state can be obtained from the method. When the above constraints are not satisfied the method is not suitable for the description of the double-exciton state.

In contrast, the SF-TDDFT approach offers a generally more flexible wavefunction description compared to TDUDFT and a reasonable description of the double-exciton state for the entire range of diradical characters, with the only inconvenience of missing some additional doubly excited configuration along with some spin complements in the wavefunction expansion, leading to spin-contaminated solutions. The SF-TDDFT method is therefore suitable to describe the double-exciton state for a more extended range of diradical systems compared to TDUDFT.

**Supplementary Materials:** The Supplementary Information are available online at http://www.mdpi.com/2079-3197/7/4/68/s1.

**Author Contributions:** F.N. conceived the research and wrote the manuscript; S.C. and Y.D. produced the data, contributed to the analysis of the results, and the writing and revision of the article.

**Funding:** This research received no external funding.

**Acknowledgments:** Support from "Valutazione della Ricerca di Ateneo" (VRA)—University of Bologna is acknowledged. Y.D. acknowledges Ministero dell'Istruzione, dell'Università e della Ricerca (MIUR) for her Ph.D. fellowship.

**Conflicts of Interest:** The authors declare no conflict of interest.

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
