# Peer review of "The Low Lying Double-Exciton State of Conjugated Diradicals: Assessment of TDUDFT and Spin-Flip TDDFT Predictions"

_computation, doi:10.3390/computation7040068_

Round 1

Reviewer 1 Report

The manuscript “The low lying double-exciton state of conjugated diradicals: assessment of TDUDFT and spin-flip TDDFT predictions”  by Canola, Dai and Negri assesses two DFT based methods to investigate the low lying double exciton state of conjugated, singlet ground state diradicals. The DFT based methods were already tested in previous work on eight experimentally characterized molecules by the same group. Here, the benchmarking is continued with four additional molecules and for one of the two methods (SF-TDDFT) an additional functional with a larger contribution of HF exchange is used. The quality of the DFT results is based on comparison to experimental values and discussed using a set of criteria (diradical character, NFOD parameter, spin contamination). It is found that BHHLYP SF-TDDFT is best suited of all tested methods to predict the energy of the double exciton state in the investigated class of molecules.

From my side, the manuscript is receiving a positive review and I have no major issues with its contents. Benchmarking studies are important to both the experts in the field of theoretical chemistry as well as the users from other areas of chemistry, thus the study has scientific value. Naturally, those kind of studies are typically not the most original or creative ones. The manuscript is mostly well written, although some points could be expressed clearer or discussed in more detail.

Most importantly, I understand from the manuscript that SF-TDDFT with BHHLYP is best suited to describe the systems, but it almost appears as if the authors somehow favor TDUDFT. For example, in lines 173 – 174 the quality of TDUB3LYP is described in a positive manner for large y0 values, even though the plot right below indicates that there is a systematic overestimation for large y0 values (which is also stated shortly after in the text). A second striking example is found when assessing the SF method (line 266 and following), where spin contamination is mentioned as one of the drawbacks of the SF method. But TDUDFT also has spin contamination. Is this effect larger in SF based methods? If so, by how much? Is it even significant? In my opinion, it should be stated more clearly that SF outperforms UDFT (or, in case this last statement is an oversimplification, it should be clarified under which circumstances TDUDFT is superior to SF). Furthermore, this finding should also be mentioned in the abstract (possibly even in the title).

A second issue concerns errors: While the systematic errors of the methods are discussed, the nonsystematic errors are not addressed. For example, Figure 4 presents rather large scatter for both methods. Are these errors significant? Could you discuss the reliability of the methods with respect to these “random” deviations?

The remaining things are mere details:

Line 13: Please explain the H,H àL,L abbreviation.

Line 25: Please write out OFETs and OPDs

Line 40: What is meant by the “location” of the excited state? (the energy?)

Caption to Figure 1: I would give the references for the molecules also here.

Lines 90 – 100: Please define NFOD in an equation as well (I overlooked the explanation of NFOD when reading the manuscript the first time).

Lines 106 – 118: This is (in part) redundant, as very similar statements were made in lines 44 – 56. Maybe you can reorganize the two paragraphs.

Figure 2. You could introduce an arrow on the left side, indicating that the diradical character increases along the y-axis. This would make the manuscript more accessible to readers from different areas of chemistry. I also think the values on the x-axis should be negative (the BS state is more stable).

Caption to Figure 3: I don’t like the term localized for the BS orbitals, as they are still delocalized over half the molecule. Maybe you can find a more accurate description (this a of course an extremely minor thing).

Line 170: The sentence starting with “The fitting line…” is unclear (an increase of underestimation). Please revise this sentence for clarity.

Line 194: Replace “liner” with “linear”.

Author Response

Reply to reviewer’s reports:

Below, for clarity, we have reproduced each comment of the reviewer in italic and followed it by the action taken.

Referee n.1

I understand from the manuscript that SF-TDDFT with BHHLYP is best suited to describe the systems, but it almost appears as if the authors somehow favor TDUDFT. For example, in lines 173 – 174 the quality of TDUB3LYP is described in a positive manner for large y0 values, even though the plot right below indicates that there is a systematic overestimation for large y0 values (which is also stated shortly after in the text).

The overestimation predicted from TDUDFT calculations, for some of the molecules featuring large y0 values, finds an explanation in the discussion section of the paper. Therefore, we have now introduced on line 166-167 the following sentence: “see the discussion section for more details”.

A second striking example is found when assessing the SF method (line 266 and following), where spin contamination is mentioned as one of the drawbacks of the SF method. But TDUDFT also has spin contamination. Is this effect larger in SF based methods? If so, by how much? Is it even significant? In my opinion, it should be stated more clearly that SF outperforms UDFT (or, in case this last statement is an oversimplification, it should be clarified under which circumstances TDUDFT is superior to SF).

The spin contamination is certainly an issue also for TDUDFT and this was, clearly, not sufficiently highlighted in the manuscript. Spin contamination is demonstrated by the analysis of the wavefunction showing (Figure 6) a large mixture with a triplet spin component for small diradical character. We have clarified this concept on lines 229-240 and 248. The text has also been amended in several parts to underscore when SF is preferable to TDUDFT.

A second issue concerns errors: While the systematic errors of the methods are discussed, the nonsystematic errors are not addressed. For example, Figure 4 presents rather large scatter for both methods. Are these errors significant? Could you discuss the reliability of the methods with respect to these “random” deviations?

The origin for the scattered results for some of the systems investigated, is analyzed in the discussion section, for TDUDFT calculations, and is attributed to a) small diradical character, b) deviation from the 2e-2o model as documented by the large NFOD value, c) excited state requiring a wavefunction in which the contribution of ground and doubly excited configurations should not be identical.  For SF-TDDFT calculations the assessment of the few scattered results is mainly attributed to a) spin contamination and b) lack of some relevant double excitation in the SF wavefunction. These concepts have been introduced on lines 271-277

Line 13: Please explain the H,H ®L,L abbreviation

We have expanded the H,H-L,L notation into HOMO,HOMO®LUMO,LUMO and introduced the short notation on line 41.

Line 25: Please write out OFETs and OPDs

We have introduced the definition for the acronyms OFET and OPD

Line 40: What is meant by the “location” of the excited state? (the energy?)

Now line 42: We have specified ‘energy’ location

Lines 90 – 100: Please define NFOD in an equation as well (I overlooked the explanation of NFOD when reading the manuscript the first time).

The equation defining NFOD has been introduced as Eq. (3)

Lines 106 – 118: This is (in part) redundant, as very similar statements were made in lines 44 – 56. Maybe you can reorganize the two paragraphs.

Although the two statements seem similar, the first is an introductory statement while the second, in the methods section, enters into the detail of the 2e-2o model and highlights the role of high level calculations to go beyond the simple model. For this reason we prefer to leave the two sentences in the two different sections.

Figure 2. You could introduce an arrow on the left side, indicating that the diradical character increases along the y-axis. This would make the manuscript more accessible to readers from different areas of chemistry. I also think the values on the x-axis should be negative (the BS state is more stable).

Following the suggestion of the referee an arrow has been introduced to indicate the increase of diradical character. The stabilization of the BS structures is reported as a positive value since it is defined in the caption as E(CS) – E(BS).

Caption to Figure 3: I don’t like the term localized for the BS orbitals, as they are still delocalized over half the molecule. Maybe you can find a more accurate description (this a of course an extremely minor thing).

We eliminated the term ‘localized’

Line 170: The sentence starting with “The fitting line…” is unclear (an increase of underestimation). Please revise this sentence for clarity.

Now line 161-162: The sentence has been rephrased.

Line 194: Replace “liner” with “linear”.

Now line 186: Corrected the missing letter

Reviewer 2 Report

The manuscript "The low lying double-exciton state of conjugated diradicals: assessment of TDUDFT and spin-flip TDDFT predictions" by Sofia Canola , Yasi Dai and Fabrizia Negri, presents a thorough study of conjugated singlet ground state diradicals for four new investigated conjugated diradicals that extends the investigation of the double-exciton state for eight diradicals previously investigated. The Authors investigated the excited state using TDDFT in the spin-flip formulation and TDUDFT theories. They stated the constraints to be satisfied for TDDFT to be valid for description of the double-exciton state, and showed that SF-TDDFT approach offers a a reasonable description of the double-exciton state although missing some spin complements in the wavefunction expansion.

The paper is nice and well written. I recommend publication in the present form.

Author Response

Referee n.2 suggested publishing the paper as it is.

Reviewer 3 Report

Negri and co-workers report a very interesting TD-DFT study about the low-lying double-exciton state of conjugated and delocalize structures, which have a strong diradical character even in the ground state. Apart from the scientific interest of these compounds for many applications (electronics, dyes…), this computational work assumes its limits (exploring multiconfigurational systems with TDDFT methods) but explores the potential use of TDUDFT and SF-TDDFT and their accuracy. It is highly relevant to be able to calculate and understand the excited states of larger systems, where multiconfigurational calculations are prohibitive from a computational cost perspective. The discussion is well written and the explanation over the relative accuracy of TDUDFT vs SF-TDDFT, depending on the value of the simple descriptor “y0,” could be helpful to choose a methodology based on a relatively “cheap” calculation. I have some small questions/comments that should be addressed, but in general, I believe this manuscript will be very interesting for the readers of Computation. Thus, I recommend publication of this manuscript after minor revisions.

Minor corrections:

Figure 1 contains different representations of the studied compounds. The authors do not clarify if they are drawing the BS Lewis structure (with radicals on it) or the fully conjugated CS Lewis structure. In most cases, the CS Lewis structure is shown (DFB, QANTHENE…) but in the case of SOZ, PT, DIPh, NZ (the new ones), there are different carbons with 3 valences (showing radical positions?). I would suggest reviewing these structures and drawing the CS Lewis structure, which is easier to understand. Alternatively, a BS Lewis structure, indicating the position of the radicals would also be acceptable. Figure 2: the R^2 value should be added to see the degree of correlation between y0 and DeltaE(BS-CS). Figure 4: The statistical analysis of both approaches (TDUB3LYP and SF-TDBHHLYP) should be added (R^2 and Mean Absolute Error). The fitting is not so good for the blue line and those parameters would help to choose the best method for a specific system. Page 6. Ln. 173-175: The authors describe the increase in error for high y0 values when TDUDFT is used. The explanation of this increment is written on page 9 ln255-259. I would recommend adding “(see discussion section for more details)” in page 6. It will help to follow the article.

Author Response

Reply to reviewer’s reports:

Below, for clarity, we have reproduced each comment of the reviewer in italic and followed it by the action taken.

Referee n.3

Figure 1 contains different representations of the studied compounds. The authors do not clarify if they are drawing the BS Lewis structure (with radicals on it) or the fully conjugated CS Lewis structure. In most cases, the CS Lewis structure is shown (DFB, QANTHENE…) but in the case of SOZ, PT, DIPh, NZ (the new ones), there are different carbons with 3 valences (showing radical positions?). I would suggest reviewing these structures and drawing the CS Lewis structure, which is easier to understand. Alternatively, a BS Lewis structure, indicating the position of the radicals would also be acceptable.

The Figure has been corrected. According to the suggestion of the referee the new set of molecules on the top part of the Figure has been redrawn as CS Lewis structures.

Figure 2: the R^2 value should be added to see the degree of correlation between y0 and DeltaE(BS-CS). Figure 4: The statistical analysis of both approaches (TDUB3LYP and SF-TDBHHLYP) should be added (R^2 and Mean Absolute Error). The fitting is not so good for the blue line and those parameters would help to choose the best method for a specific system.

The R squared value has been included in the caption to Figure 2. The statistical analysis has not been included in the case of Figure 4 because of the qualitative value of the linear fitting. A new sentence is introduced on line 159 to highlight this. Deviations are indeed due to the combination of different factors analyzed in the discussion section and that we have further clarified in lines 271-277

Page 6. Ln. 173-175: The authors describe the increase in error for high y0 values when TDUDFT is used. The explanation of this increment is written on page 9 ln255-259. I would recommend adding “(see discussion section for more details)” in page 6. It will help to follow the article.

Now  lines 167-168: following the suggestion of the referee we have introduced “(see the discussion section for more details)”
